



# Hydrometeorological drivers of flood characteristics in the Brahmaputra river basin in Bangladesh

Sazzad Hossain[1,2], Hannah L. Cloke[1,3,4,5], Andrea Ficchì[1], Andrew G. Turner[3,6], Elisabeth M. Stephens[1]

[1]Department of Geography and Environmental Science, University of Reading, Reading, UK
[2]Flood Forecasting and Warning Centre, BWDB, Dhaka, Bangladesh
[3]Department of Meteorology, University of Reading, Reading, UK
[4]Department of Earth Sciences, Uppsala University, Uppsala, Sweden
[5]Centre of Natural Hazards and Disaster Science, CNDS, Uppsala, Sweden
[6]National Centre for Atmospheric Science, University of Reading, Reading, UK

*Correspondence to:* Sazzad Hossain (mdsazzad.hossain@pgr.reading.ac.uk) and Elisabeth M. Stephens (elisabeth.stephens@reading.ac.uk)

**Abstract.** While flooding is an annual occurrence in the Brahmaputra basin during the South Asian summer monsoon, there is large variability in the flood characteristics that drive risk: flood duration, rate of water level rise and peak water level. The aim of this study is to understand the key hydrometeorological drivers influencing these flood characteristics. We analyse hydrometeorological time series of the last 33 years to understand flood dynamics
focusing on three extraordinary floods in 1998 (long duration), 2017 (rapid rise) and 2019 (high water level). We find that long duration floods in the basin have been driven by basin-wide seasonal rainfall extremes associated with the development phase of strong La Niña events, whereas floods with a rapid rate of rise have been driven by more localized rainfall falling in a hydrological 'sweet spot' that leads to a concurrent contribution from the tributaries into the main stem of the river. We find that recent record high water levels are not coincident with extreme river
flows, hinting that sedimentation and morphological changes are also important drivers of flood risk that should be further investigated. Understanding these drivers is essential for flood forecasting and early warning and also to study the impact of future climate change on flood.


## 1 Introduction

The most impactful floods in Bangladesh occurred in 1998, affecting around 31 million people. Their long duration caused crop damage across 1.5 million hectares, 4,500 km of embankments and 16,000 km of roads were also
damaged (Islam, 2000) and there were economic losses of up to US$ 4.3 billion in 1998 (EM-DAT) (see supplementary Fig. S1). More recently, 8.5 and 6.5 million people were affected during the 2017 and 2019 floods respectively (DDM, 2017, 2019). Understanding the drivers of floods is important for flood forecasting and early warning (Stephens et al., 2015) due to increased vulnerability from transboundary floods (Bakker, 2009). Bangladesh lies at the lower riparian part of three large transboundary basins: the Ganges, Brahmaputra and
Meghna basins (Fig. 1.), with the Brahmaputra having the most widespread and impactful flooding. In a normal monsoon year around 20% of the country is affected by floods (Rahman et al., 2013) and in extreme flooding years this rises to more than 50% (Mirza, 2003). This flooding causes enormous damage to crops and physical infrastructure such as houses, roads and flood defences. The extent of damage is dependent on the timing, duration, and extent of the flood pulses during the monsoon season (June–September) which vary considerably
among the different years (Fig. 3), and the rate of rise, which influences the ability to provide sufficient early warning.





Understanding the major drivers of these floods is key to informing the development of reliable early warning systems (Blöschl et al., 2013; WMO, 2013) and accurate predictions of future flood hazard in a changing climate (Merz et al., 2012). However, there is currently a lack of understanding of how interannual and intraseasonal monsoon variability affects the flood characteristics which drive risk in the Brahmaputra, despite indications that
this variability has important controls on the river discharge (Jian et al., 2009). During a developing La Niña event, an enhanced Walker circulation over the maritime continent drives intensified precipitation during the Indian Summer monsoon (Shi and Wang, 2019). However, the influence of La Niña has been proposed to be diminishing in recent years due to weaker La Niña events and the impact of warmer Indian Ocean temperature (Samanta et al., 2020).  Analysis of the floods at seasonal or monthly scales (Islam and Chowdhury, 2002; Islam et al., 2010;
Mirza, 2003) is inadequate for the investigation of the drivers of flood events (Gaál et al., 2012), especially for floods which have a rapid rise in water level (e.g. 2017 floods); events for which the Bangladesh Flood Forecasting and Warning Centre (FFWC) is keen to improve their early warnings.

In order to determine the hydrometeorological drivers of flood characteristics in the Brahmaputra, we analyse antecedent conditions (Blöschl et al., 2013; Carter and Steinschneider, 2018; Schröter et al., 2015), large-scale
atmospheric and ocean anomalies (Paeth et al., 2011) and extreme statistics of precipitation and river flow (Schröter et al., 2015). We analyse the three most severe examples of flood events with different characteristics of the Brahmaputra river: (i) long duration flooding in 1998, (ii) flooding with a rapid rise in water levels in 2017 and (iii) flooding with high water levels in 2019.

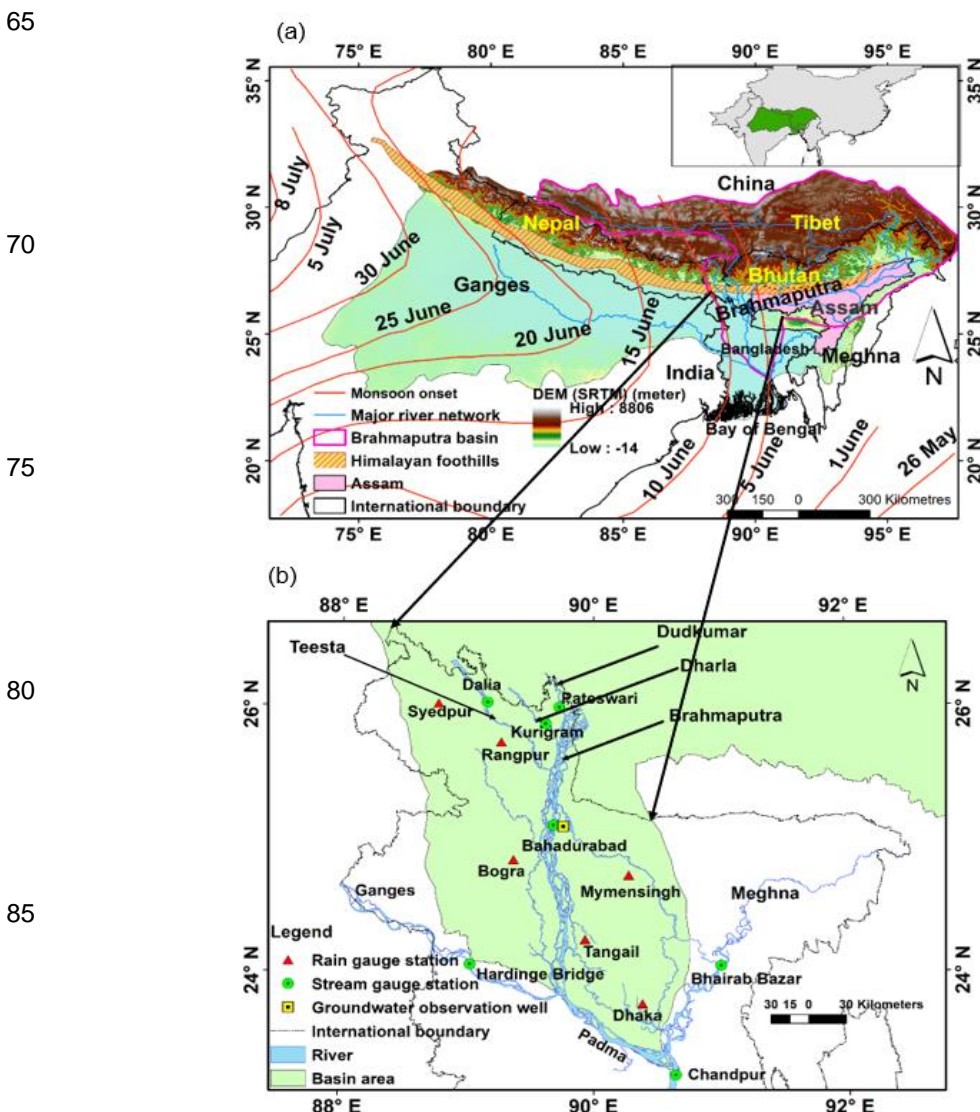

**Figure 1.** (a) Location of the Brahmaputra basin in South Asia with respect to the Ganges and Meghna basins. The Himalayan foothills and Assam areas are highlighted. Monsoon onset dates are shown with red lines (Pai et al., 2020). (b) Basin area in Bangladesh with major river systems. The locations of the stream gauges, rain gauge stations and groundwater observation well are also shown (stations position from the FFWC).

## 2 Basin characteristics

### 2.1 Geography

The Brahmaputra basin is located between approximately 82° E to 97° 50´ E and 25° 10´ N to 30° 30´ N with a total area of about 580,000 km² (Bora, 2004). It is a transboundary river basin shared by Bangladesh (8.1%), Bhutan





(7.8%), China (50.5%) and India (33.6%) (Goswami and Das, 2002).The river length is about 2880 km of which 1625 km is in Tibet (China), 918 km in India and 337 km in Bangladesh (Sarma, 2005). The river meets with the Ganges at Goalondo in Bangladesh where it becomes known as the Padma. It finally meets with the Meghna river at Chandpur

before flowing into the Bay of Bengal (Fig.1). The Teesta, Dharla and Dudkumar are three major tributaries inside Bangladesh. The physiography of the basin can be classified into three distinct zones: the Tibetan plateau (elevation exceeding 3500 m); the Himalaya belt (elevation between 100 m and 3500 m) and the agricultural flood plain (elevation up to 100 m) (Immerzeel, 2008). The river is braided, and its width varies from 9 to 16 km during the monsoon in Bangladesh (BWDB, 2011).


**2.2 Hydrometeorology and flood characteristics**

The Brahmaputra basin receives between 60–70% of its annual rainfall from June–September, with 20–25 % falling during the pre-monsoon of April and May (Dhar and Nandargi, 2000; Purkait, 2004). The spatial distribution of

rainfall in pre-monsoon and monsoon seasons (Fig. 2a and 2b, respectively) means that daily rainfall in the southeast of the basin (Bangladesh and Assam) is up to 10 times higher than in the northwest. The pre-monsoon rainfall is primarily caused by thunderstorm activity and movement of depressions from the west towards the basin (Khatun et al., 2016; Purkait, 2004). The onset of the monsoon precipitation usually occurs in the first week of June (Fig. 1a) and rainfall follows in a sequence of wet and dry spells. Monsoon heavy rainfall in the Brahmaputra basin

is associated with the movement of the eastern end of the monsoon trough to the Assam region, producing break monsoon conditions over central India and active conditions around the Himalayan foothills (Fig. S2), often with a monsoon depression originating from the Bay of Bengal, which recurves northwards over the Brahmaputra basin i.e. Bangladesh and Assam (Dhar and Nandargi, 2003; Dhar and Nandargi, 2000; Nandargi and Dhar, 2011). The annual mean rainfall in the Arunachal Pradesh, Assam and sub-Himalayan regions is about 2300 mm with some

foothill regions receiving as much as 5000 mm (Dhar and Nandargi, 2000; Singh et al., 2013). The high altitude Tibet region of the Brahmaputra has an annual mean rainfall of ~730 mm (Immerzeel, 2008). The daily evapotranspiration (ET) is 3–4 mm per day and monthly ET is comparable to (or slightly exceeds) the rainfall in pre-monsoon months when the temperature is high (Fig. 2c). During the monsoon, rainfall always exceeds ET, leading to a rise of river water level .

The Brahmaputra is partly fed by upstream glacial snowmelt (Immerzeel, 2008; Masood and Takeuchi, 2015; Paura, 2004) which contributes to the river baseflow. Both river and groundwater levels start to rise in April and peak between July and August (Fig. 2d) in response to the monsoon rainfall. The groundwater is also recharged annually by the monsoon precipitation, which contributes significantly to baseflow. There is large spatial variability in maximum daily river flows: at Tsela Dzong in the Tibetan plateau, the flow is approximately 10,200 $m^3 s^{-1}$, which

is only 14 % of the maximum daily flow recorded at Guwahati in Assam (draining approximately 73 % of the catchment) and 10 % of the maximum daily flow recorded at Bahadurabad in the river delta in Bangladesh (Datta and Singh, 2004) (see Fig. 1b for location). It is the monsoon rainfall received in the high precipitation recipient zones of Himalayan foothills and the southern floodplain region that controls the high flows of the river downstream in Assam and floodplain delta region in Bangladesh. Flooding in Bangladesh is categorised by the FFWC using a

'danger level': the threshold at which water starts to cause damage to property, crops or other infrastructure. A 'flooding situation' is identified when water levels cross the danger level, while a 'severe flooding' situation is when water levels exceed the danger level by 1 m or more.



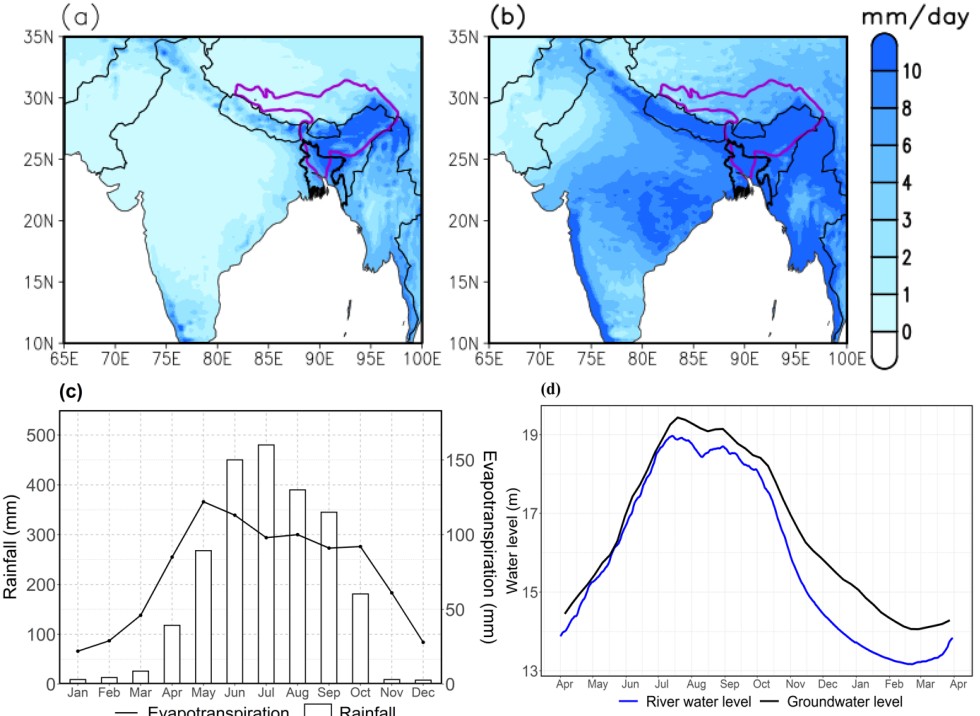

**Figure 2.** Climatology of mean rainfall (mm day⁻¹) in (a) pre-monsoon months, April-May, (b) monsoon season, June-September, based on ERA5 reanalysis (period:1987–2016). The basin boundary is shown in purple. (c) Monthly rainfall (mm) and evapotranspiration, ET, (mm) at Rangpur (source: Bangladesh Meteorological Department (BMD)). (d) Daily average river water level and groundwater level (m) at Bahadurabad starting from April based on 1987–2016 (see Figure 1b for location of the stations) (source: Bangladesh Water Development Board (BWDB)).

Flood characteristics (magnitude, timing, duration, and number of events) vary each year in response to variations in the monsoon rainfall (Fig. 3). On average, moderate floods occur once in 2 years, while severe floods occur once in 6–7 years (Siddique and Chowdhury, 2000). Over the last 33 years, 'severe flooding' has occurred during the 1988, 1998, 2012, 2016, 2017 and 2019 monsoons (FFWC, 2019). In this work we link the characteristics of each flood to the meteorological and hydrological drivers which play a key role in shaping flood variations annually.





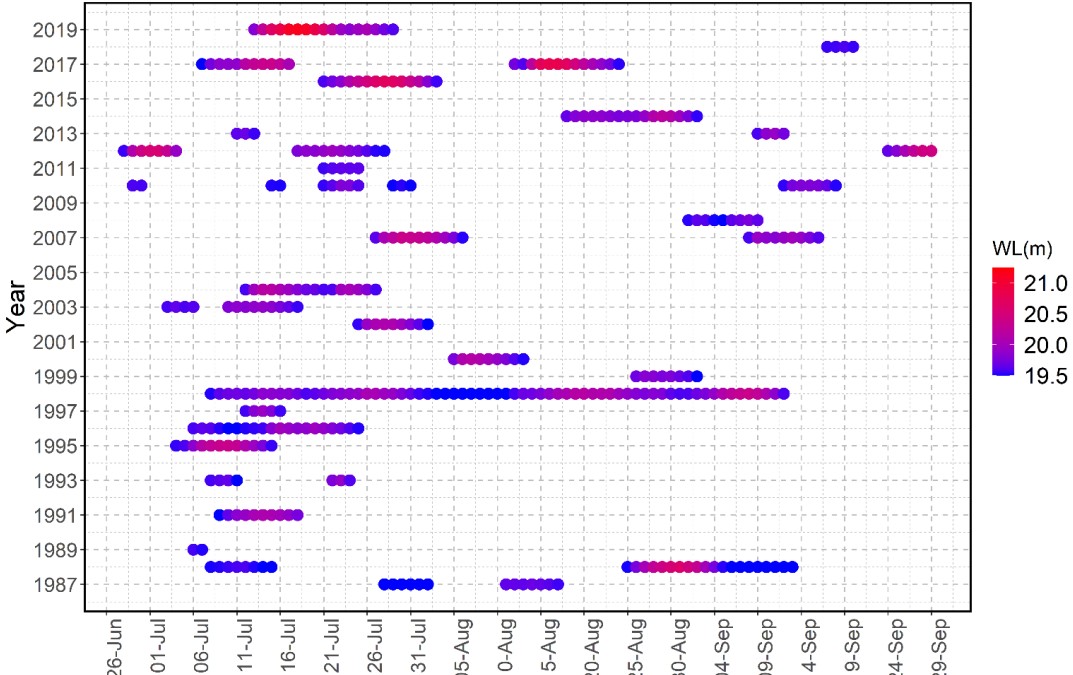

**Figure 3.** Dates indicated by a coloured dot show when the water level (WL) exceeded the danger level at the Bahadurabad station on the Brahmaputra river (see Fig.1b for location of the gauge). The colour indicates the water level (from low, blue, to high, red).

## 3. Data

### 3.1 Observed data

Observed daily water level and river flow data are provided by the hydrological division of the Bangladesh Water Development Board (BWDB) for the 33-year period from 1987 to 2019. Four key flood monitoring river gauges have been used in this study: Bahadurabad (Brahmaputra), Dalia (Teesta), Kurigram (Dharla) and Pateswari (Dudkukmar) (Fig. 1b). In addition, water level data from the Ganges and the Meghna has been used to demonstrate the flood synchronization of these two large rivers with the Brahmaputra. Water level data is collected using a manual water level staff gauge at 3-hour intervals five times per day between 6:00 AM and 6:00 PM local time, with no data at night. The river flow is measured using a current meter (or Acoustic Doppler Current profiler) approximately twice a month to compare with water level, and a continuous time series of daily river flow is estimated based on the stage-discharge relationship for the same length of record as the water level data. Observed rainfall data from the Bangladesh Meteorological Department (BMD) has been used to study the rainfall events over the same period as hydrological data. There are six rain gauge stations located inside the Brahmaputra basin in Bangladesh (Fig. 1b).



**3.2 Reanalysis data**

High resolution daily gridded precipitation and soil moisture data of ERA5 reanalysis (Hersbach et al., 2020) have been used in order to study the large-scale rainfall situation and soil saturation. Data was retrieved from the Climate Data Store (CDS) of the Copernicus Climate Change Service for the period 1987–2019. The output resolution of ERA5 is 0.25 degree with global coverage and hourly time samples. The ERA5 reanalysis has been determined to

the most suitable reanalysis for hydrological applications in the Indian monsoon region (Mahto and Mishra, 2019).

**3.3 Climate indices data**

Large-scale climate indices have been used to study ENSO and tropical intra-seasonal oscillations (ISOs), namely

the boreal summer intra-seasonal oscillation (BSISO) for different flooding years. To classify El Ninõ (La Ninã) years, we use monthly SST anomaly based on the monthly Extended Reconstructed Sea Surface Temperature (Huang et al., 2017) for the Niño 3.4 region (5° N to 5° S, 170° W to 120° W) available from NOAA (2020) for the period 1987–2019.

We use the ISO indices of Kikuchi et al. (2012) to represent the phase and amplitude (location and strength) of the

BSISO that are available from IPRC/SOEST (2020) for the same period as ENSO. The ISO indices are derived based on the first two principle components (PC1 and PC2) of extended empirical orthogonal functions (EEOFs) of outgoing longwave radiation (OLR) with a 25–90 days filtered time series over the tropics between 30° S to 30° N (Kikuchi et al., 2012). The first two PCs (PC1 and PC2) for each mode are used to determine the strength and phase of the BSISO (Kikuchi et al., 2012).


**4. Methods**

**4.1 Meteorological drivers**

**4.1.1 Large-scale atmospheric drivers**

The ENSO state is known to influence the interannual variability of monsoon rainfall (Krishnamurthy and Kinter, 2003; Nanjundiah et al., 2013). During a developing La Niña (El Niño), the monsoon strengthens (weakens) compared to ENSO neutral years (Samanta et al., 2020; Webster et al., 1998). ENSO begins in boreal spring (March–April), and usually peaks at the end of the year, decaying during the following spring. We used the SST

anomaly for the Niño 3.4 region averaged over November–January (NDJ), and classified ENSO years based on a comparison with the standard deviation of the anomaly (>1 = strong ENSO, 0.5 –1.0 = weak ENSO,<0.5 neutral conditions) (Santoso et al., 2017). ENSO events are also classified by whether they are developing or decaying years if the event spans multiple years. From this classification we calculated the composite of June–September rainfall to compare between strong developing La Niña years and other years.

The intra-seasonal variation of monsoon rainfall is marked by wet and dry spells known as active and break events, with typical lifespans of around two weeks (Krishnamurthy and Shukla, 2007). There are two dominant modes of tropical ISOs linked to the intra-seasonal variability of the monsoon (Lee et al., 2013): the Madden-Julian Oscillation MJO is dominant during boreal winter with eastward propagation along the equator (Kikuchi et al., 2012; Lee et al.,





2013), whereas active and break events form part of the 30-50 day intra-seasonal variation as part of the BSISO,
featuring northward-propagating bands of convection at South Asian longitudes together with eastward propagation
along the equator, akin to the MJO (Annamalai and Sperber, 2005). The MJO mode dominates from December to
April (Kikuchi et al., 2012) so, is not considered as an important driver for the Brahmaputra floods, which occur
between June and September.

BSISO events are identified at a particular time during the monsoon if the amplitude $(amplitude = \sqrt{PC_1^2 + PC_2^2})$
is greater than (or equal to) 1, whereas amplitudes less than 1 are considered as weak conditions (Kikuchi et al.,
2012). The phase space diagram (8 phases, a variant of that used to classify the MJO as in (Wheeler and Hendon,
2004)) shows the advancement of BSISO, which originates in the equatorial Indian Ocean and propagates in a
northwards direction (Kikuchi et al., 2012). We calculated the climatology of the rainfall anomaly for the 8  phases
of  BSIOS events (amplitude >1) including weak phase when amplitude <1 irrespective of BSIOS phase and study
the phase-space diagram for the 1998, 2017 and 2019 monsoons.

### 4.1.2 Rainfall characteristics

The spatial and temporal variation of monsoon rainfall was analysed based on the magnitude, intensity, duration
and spatial distribution of rainfall events, as well as monthly anomalies and accumulations over the monsoon period
(June–September). Rainfall extremes were analysed by developing a depth-duration frequency curve using the
Generalized Extreme Value (GEV) method.

Rainfall events were defined using the method to identify 'wet spells' described in Singh and Ranade (2010), which
identifies continuous periods with daily rainfall equal to or greater than the daily mean rainfall of climatological
monsoon period using the following five steps:

(1) computation of daily rainfall climatology;

(2) calculation of daily mean rainfall (mm/day) over the summer monsoon period (June–September) over all years
(climatology);

(3) normalization of year-wise daily rainfall by dividing by the daily mean monsoon rainfall;

(4) smoothing normalised daily rainfall time series with a 9-point Gaussian low-pass filter;

(5) identification of wet and dry spells as continuous smoothed daily rainfall >1 and <1 respectively.

### 4.2 Hydrological drivers

### 4.2.1 Analysis of hydrological time series

In this study, the 1-D discrete wavelet transform (DWT) was used to decompose river water level and discharge data
to identify short-term variations and the annual cycle. This approach allows a clear comparison across years, as it
gives an indication of the relative importance of the seasonal hydrological cycle compared to specific rainfall events
for different flooding years. The Daubechies wavelet function is commonly used in hydrometeorological time series
analysis (Chen et al., 2016; Franco Villoria et al., 2012; Pandey et al., 2017; Zhang et al., 2017). It was used here to
decompose the daily water level and river flow time series from 1987 to 2019 into 6 detailed components (D1-D6) and
one approximation (A6). The detailed components present the variation in $2^n$ (dyadic translation) where n is the level





of the detailed component. The daily time series D1 to D6 therefore represent 2-day, 4-day, 8-day, 16-day, 32-day and 64-day periodicity respectively.

We also performed a trend analysis on time series of annual maxima of water level and river flow to investigate possible trends. The significance of the trends was assessed by using the Mann–Kendall test.

### 4.2.2 Hydrological characteristics


We analysed three end-members of different flood characteristics of the Brahmaputra: (i) long duration flooding, such as the flooding that occurred in 1998; (ii) flooding with a rapid rise in water levels, such as the flooding that occurred in 2017 and (iii) flooding with high water levels, such as the flooding that occurred in 2019. Hydrological characteristics were studied in terms of initial water level, rate of rise of the water level, duration, annual peak water

level, and synchronisation of floods. The exceedance probability of annual maximum discharge of the Brahmaputra river was calculated using the GEV distribution for both river flow and water level. This allows a discussion of the 1998, 2017 and 2019 floods in comparison to other years.

### 4.2.3 Soil moisture evolution


Soil moisture evolution and seasonal soil moisture saturation anomalies were studied for the upper layer (0 to 7 cm) based on ERA5 volumetric soil moisture. The percentage saturation was calculated using soil moisture at saturation level and respective residual moisture from the ECMWF's land surface model (HTESSEL) used in the Integrated Forecast System (ECMWF, 2018).


## 5 Results

### 5.1 Meteorological drivers


### 5.1.1 Large scale drivers

The Brahmaputra basin receives more rainfall in La Niña conditions, as shown by the positive anomaly of the seasonal (June–September) rainfall for strong La Niña developing years (1988,1998, 2007, 2010) compared to all

others (Fig. 4a). ENSO classification based on the SST anomaly at Niño 3.4 region is shown in Fig. 4b. During 1988, 1998, 2007 and 2010, seasonal rainfall was 17 %, 20 %, 8 % and 10 % higher than long-term average (1987–2016). Rainfall in the basin during the 2017 weak La Niña and 2019 neutral conditions was similar to the long-term average. We find that two long duration floods in 1998 and 1988 occurred in strong La Niña developing years. The flood durations in 1988 and 1998 were 27 and 66 days respectively, while in the other two strong La Niña years

(2007 and 2010), flood durations were 20 and 19 days. In strong El Niño years, we find 2009 to be a dry monsoon year for the basin with no floods and the basin received 9 % less rainfall than the long-term average. However, 1997 and 2015 (strong El Niño) were normal monsoon years for the Brahmaputra basin with 7 and 20 days flood durations in 1997 and 2015 respectively; for weak La Niña and neutral conditions during 2017 and 2019, flood durations were 25 and 17 days respectively.



We find limited evidence for a relationship between the BSISO mode of intraseasonal oscillation and flooding in the Brahmaputra basin. The position and amplitude of BSISO is recorded in 8 phases starting from the equatorial Indian Ocean and moving northward to the Bay of Bengal (Kikuchi et al., 2012), and an amplitude greater than 1 in phases 1 to 3 is associated with enhanced rainfall in the Brahmaputra basin (Fig. S3). From here on we refer to this condition as a 'BSISO event'. The average total duration of BSISO events (i.e., number of days where it is

'strong' in phases 1–3 during the monsoon season) is 25 days (averaged over 1987–2019). During the 1998, 2017 and 2019 floods the duration of BSISO events was 14, 12 and 24 days respectively (Fig. 5). In July 1998 rainfall events occurred during BSISO events for 14 days but in August it remained almost solely in a weak state, despite the record long duration floods. The heavy rainfall event/floods in August 2017 was associated with a strong BSISO event, but the rainfall/flood events in July 2019 occurred during the weak phases  i.e. amplitude  less than 1 in

phases 1 to 3  of the BSISO (Fig. 5), and there was no flooding while it was active for 24 days in August and September.

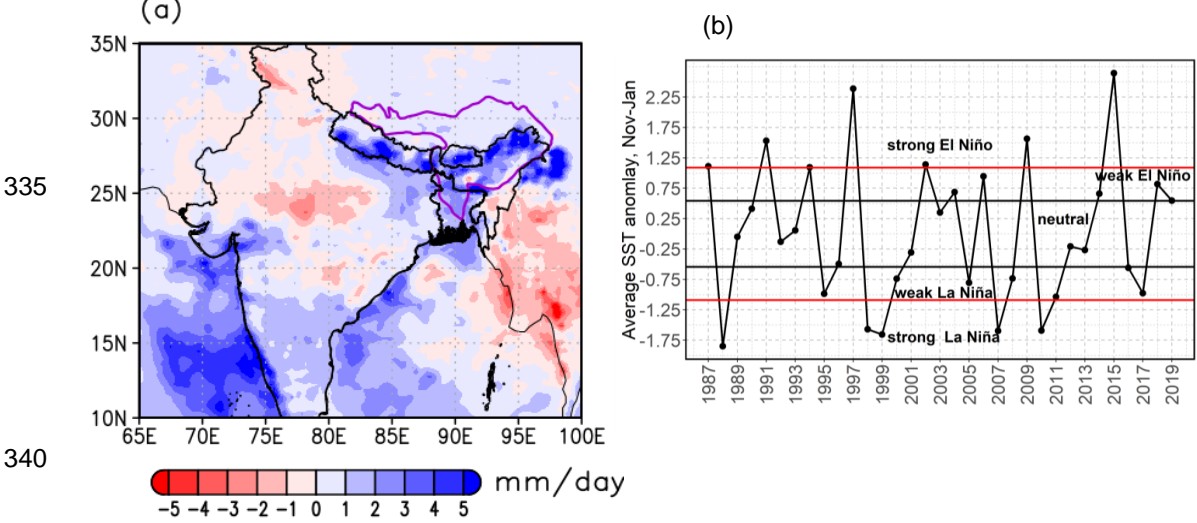

**Figure 4.** (a) Mean JJAS rainfall (mm day⁻¹) difference between strong La Niña development years (1988,1998 and 2007) and all other years over 1987–2019 (based on ERA5 reanalysis); (b) Classification of ENSO years as strong, neutral and weak based on SST anomalies of November to January in the Niño 3.4 region (5° N–5° S, 120° W–170° W) and horizontal red and black line 1 and 0.5 standard deviations (NOAA, 2020).

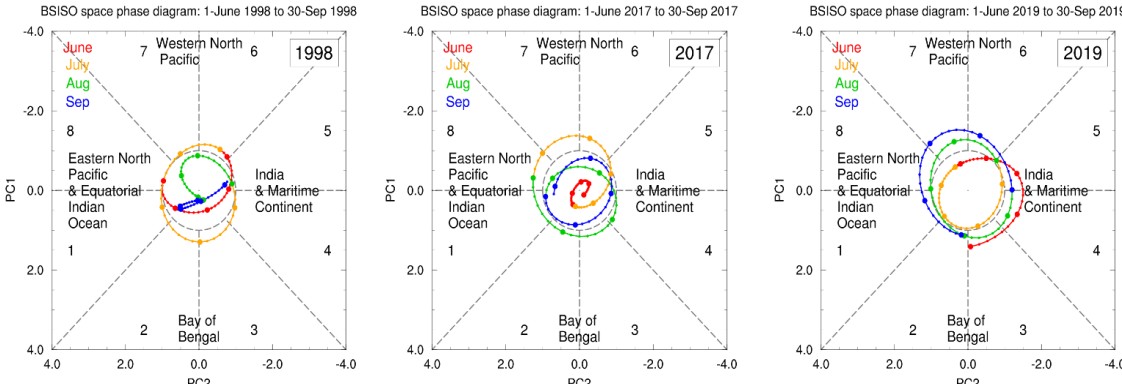

**Figure 5.** Phase-space diagram of BSISO index for the three monsoon (June–September) years.





### 5.1.2 Monsoon rainfall events

Rainfall events vary substantially across the monsoon period, and this plays a key role in triggering floods with
different characteristics. The number of rainfall events in 1998 (6), 2017 (7) and 2019 (5) were not exceptional;
however, the duration, intensity, amount and spatial distribution of rainfall varies substantially among the different
years (Table 1). The spatial distribution of monsoon rainfall events varies in homogeneity and coverage of the basin
prior to the flood peak: in 1998 rainfall events extended across the basin for a longer duration and were more
coincident with floods; in 2017 (7 to 13 August) rainfall was concentrated in small areas of the lower sub-basins
(Teesta, Dharla and Dudkumar) located in Jalpaiguri, Cooch Behar, Bhutan and Bangladesh, forming a hydrological
sweet spot; in July 2019 the rainfall distribution was relatively widespread across the Brahmaputra basin over an
extended period (Fig. 6).

For the long duration floods in 1998, the annual exceedance probability (AEP) of individual rainfall events was not
as extreme as those in 2017 and 2019, with the rapid rate of rise in 2017 driven by a particularly extreme event of
7 days /169 mm (24 mm day$^{-1}$) / 20 % AEP; with high water level floods in 2019 driven by a particularly extreme
event of 12 days / 300 mm (23 mm day$^{-1}$) / 4 % AEP (Table 1). However, the total seasonal rainfall was more
extreme for the long duration floods in 1998 (1% AEP) than these relatively short duration floods in 2017 (29 %
AEP), 2019 (67 % AEP) (Table 1 and Fig. S4).

**Table 1.** Monsoon rainfall events average over the Brahmaputra basin (based on ERA5 reanalysis)

| Year | Rainfall events | Event accumulated rainfall (mm) | Annual exceedance Probability (AEP) | Average rainfall intensity during event (mm/day) | Seasonal total (mm) (AEP) | Remarks |
|---|---|---|---|---|---|---|
| 1998 | 19 June to 24 June | 83 | 1 | 14 | 1559 (0.01) | |
| | 4 July to 25 July | 372 | 0.33 | 17 | | Co-occurring flood event |
| | 30 July to 5 August | 112 | 0.67 | 16 | | Co-occurring flood event |
| | 10 August to 21 Aug. | 199 | 1 | 17 | | Co-occurring flood event |
| | 26 August to 5 September | 197 | 0.50 | 18 | | Co-occurring flood event |
| | 17 Sep to 21 Sep | 35 | 1 | 7 | | |
| | **Number of events: 6** | **Total: 998** | | **Average: 15** | | |
| 2017 | 8 June to 19 June | 147 | 1 | 12 | 1331 (0.29) | |
| | 29 June to 12 July | 250 | 0.40 | 18 | | Co-occurring flood event |





| | | | | | | |
|---|---|---|---|---|---|---|
| | 20 July to 1 August | 135 | 1 | 10 | | |
| | 7 August to 13 August | 169 | 0.20 | 24 | | Co-occurring flood event |
| | 22 August to 28 August | 48 | 1 | 7 | | Co-occurring flood event |
| | 16 Sep to 21 Sep | 45 | 1 | 8 | | |
| | 23 Sep to 30 Sep | 61 | 1 | 8 | | |
| | **Number of events: 7** | **Total: 855** | | **Average: 12** | | |
| 2019 | 25 to 28 June | 68 | 1 | 17 | 1276 (0.67) | |
| | 4 to 16 July | 300 | 0.04 | 23 | | Co-occurring flood event |
| | 22 to 26 July | 78 | 1 | 16 | | Co-occurring flood event |
| | 9 August to 16 August | 76 | 1 | 10 | | |
| | 9 Sep to 15 Sep | 94 | 1 | 13 | | |
| | Number of events:5 | **Total: 616** | | **Average: 16** | | |












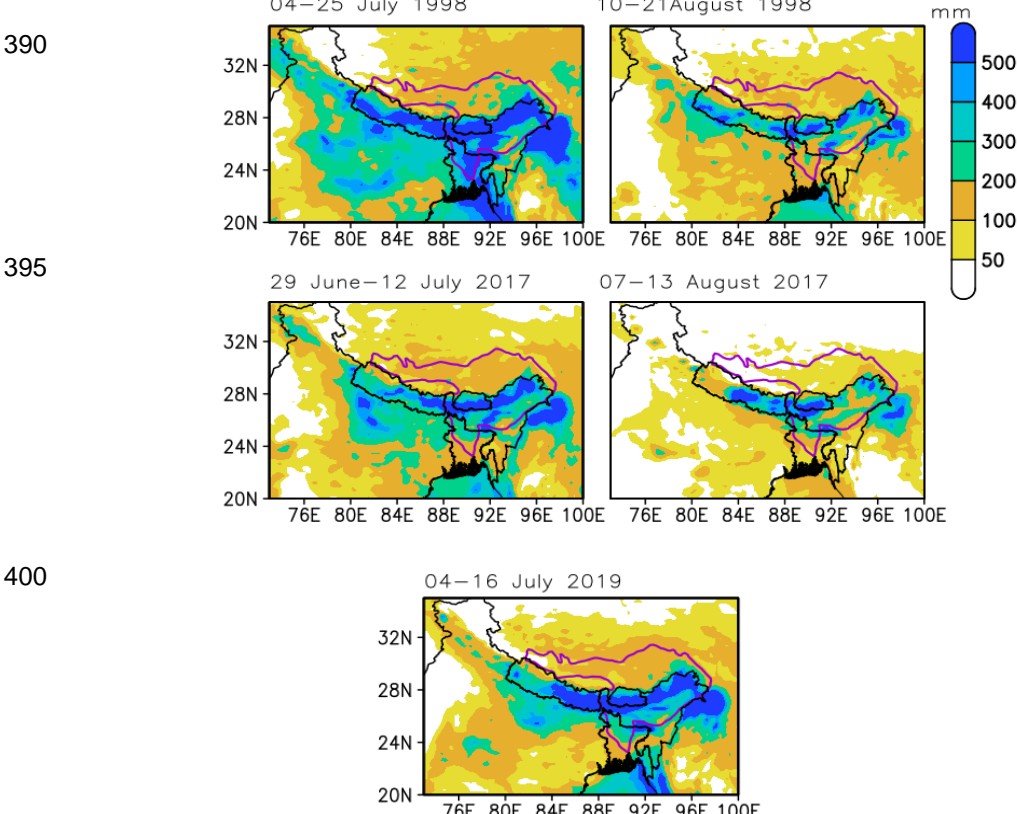

**Figure 6.** Spatial distribution of flood-triggering rainfall events in 1998, 2017 and 2019 (Source: ERA5 reanalysis).

### 5.1.3 Monthly rainfall anomalies

In 1998, monthly rainfall totals in June, July and August were 18%, 22% and 48% above normal respectively, and the pattern was (almost) basin-wide (Fig. 7a). In 2017, the first two monsoon months (June–July) of the season were almost normal, i.e., June and July only 1.2% and 3.4% less respectively, but flooding occurred after strong positive rainfall anomalies in August 2017 in the lower part of the basin, especially in Bangladesh and adjacent areas (Fig. 7b). The basin received 14% more than its long-term average in August. In 2019, June (26 % less) and August (27 % less) were much drier than normal over the whole basin (Fig. 7c) but the above-normal rainfall across the whole Brahmaputra basin in July (27% more) caused flooding. During the 2019 monsoon, the basin received a single flood wave only in July and no floods in August, as there was no remarkable rainfall event (Table 1) and the monthly anomaly was drier than normal (Fig. 7c).

Monsoon cumulative rainfall was higher than the climatological average over the basin for the long-duration floods in 1998, with the rate of accumulation steady throughout the period (Fig. 7d). In 2017 the rainfall accumulation was much less steady, with a sudden jump visible in the curve in early August. Such an abrupt step-increase in rainfall rates led to a rapid rise of the river. In contrast to 1998 and 2017, the 2019 monsoon rainfall was below the



climatological average as there were no remarkable high rainfall events except for the period between mid-July to the first week of August.

**Figure 7.** Monthly rainfall anomalies in (a) 1998; (b) 2017; (c) 2019 and (d) Cumulative rainfall (over the basin) from June to September (based on ERA5 reanalysis).


## 5.2 Hydrological drivers

### 5.2.1 Annual cycle and sub-seasonal variability of flooding

430

The annual evolution of the Brahmaputra river has both low and high-frequency components, representing a unimodal annual cycle with strong sub-seasonal variability on time scales of a few weeks. The water level and river flow hydrograph of the Brahmaputra river have been decomposed into high-frequency (variability within the annual hydrological cycle) and low-frequency components (residual hydrograph where the high-frequency components have been removed, i.e. seasonal regime or trend) to investigate the variability during the monsoon floods in different years. We used the wavelet analysis methods (see Section 4.2.1) and we selected the 16-day (D4) component of periodicity, which is the dominant high-frequency component that aligns with the typical time scales of wet and dry spells. The hydrological regime has a distinct unimodal annual cycle with the annual peak in water level / river flow between July and August (Fig. 8b and 8e). The flood water usually starts to recede from the end of August, however in some years flooding continues until the second week of September (Fig. 8a and 8d). The hydrological time series show strong sub-seasonal variability within the annual cycle (Fig. 8c and 8f).

While the 1998 floods show a less pronounced short-term (high-frequency) variability (Fig. 8c and 8f), the seasonal (low frequency) component was stronger than all other years for both river flow and water level (Fig. 8b and 8e). For the 2017 floods, the seasonal regime (general trend, A6) was not exceptional (Fig. 8b), but in August the high-frequency component (short-term variability) reached the largest amplitude among all years (Fig. 8c and 8f). On the other hand, during the 2019 monsoon, though the water level exceeded all previous historical recorded levels (Fig. 8a) the high and low-frequency components were not as strong as the 1998 or 2017 floods. There was a synchronisation in the peaks of the low and high-frequency components, which may be important given that they were lower than in 1998 and 2017 respectively. In addition, the recorded high-water level in 2019 is not matched by the river flow, which suggests that other drivers other than the hydrometeorology are also important.

455

460

465

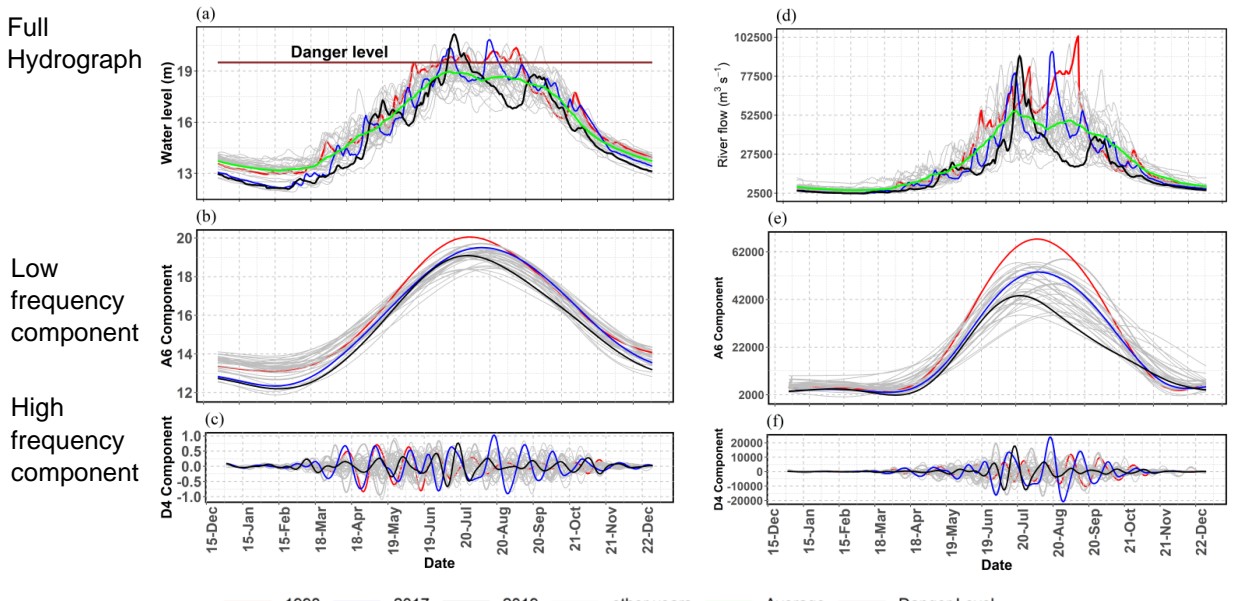

**Figure 8.** Left: (a) Full annual hydrograph from observed daily water level (m) at Bahadurabad gauge station (data from 1987 to 2019), (b) Low frequency component (A6) of wavelet transform, with a six-level decomposition of daily water level and (c) High frequency component (D4) of wavelet transform at 16-days variation. Similarly, right panels at Bahadurabad (d, e and f) show river flow ($m^3 s^{-1}$) hydrograph, low frequency component and high frequency component respectively.

### 5.2.2 Antecedent water level

While pre-monsoon (April–May) water levels were higher in 1998 compared to 2017 and 2019, they were still slightly below the long-term average (Fig. 8a). In 2017 and 2019 the water levels during the monsoon season fluctuated around the long-term average, whereas in 1998 water levels stayed well above the long-term average from mid-June to mid-September. This was driven by above-average precipitation throughout the monsoon season (Fig. 7a). Thus, the high initial water level played a role for the floods in 1998, whereas in 2017 and 2019, floods occurred starting from water levels below normal even a few weeks before the flood event. Therefore, intense short-duration rainfall events just before the floods primarily caused flooding in 2017 and 2019.

### 5.2.3 Soil moisture

The soil moisture starts to increase gradually with the contribution of the pre-monsoon rainfall during April to May and reaches its annual maximum level with the monsoon rainfall in June (Fig. 9a). During the monsoon soil moisture is provided by successive rainfall events, and it is likely that basin-wide and frequent rainfall events maintained soil moisture above the average for the long duration floods in 1998. In 2017 and 2019 soil moisture fluctuated around this long-term average throughout the monsoon season. The basin-wide soil moisture anomalies for the entire monsoon season are quite similar for 1998 and 2017, however, for the high water level floods during the 2019 monsoon, soil moisture anomalies were negative across large parts of the basin (Fig. 9b) because the basin received below-normal rainfall in both June and August (as shown in Fig. 7c).





495

**Figure 9.** (a) Annual evolution of soil moisture (topsoil soil layer, 0–7 cm) averaged over the basin for the period from 1987 to 2019. (b) Soil moisture anomaly during monsoon season (June-September) in 1998, 2017 and 2019 (based on ERA5 reanalysis).

### 5.2.4 Peak water level and discharge

The peak water level was the most striking feature in the 2019 flood as the Brahmaputra, along with its tributary Teesta recorded the highest annual maximum water level than their previous records while the river discharge was not as high compared to some other previous floods. An almost similar behaviour of water level was recorded during the 2017 floods (Fig. S5). The water level of the annual maximum peak of the Brahmaputra at Bahadurabad stream gauging station in 1998, 2017 and 2019 was respectively 87 cm, 134 cm and 166 cm above the danger level, which correspond to annual exceedance probabilities of 10.77 %, 3.08 %, 1.54 % respectively (Fig. 10a). When the measured discharge is not available on the day when the water level is maximum in the river, it is estimated using a rating curve. The measured discharge (water level) of the Brahmaputra river at Bahadurabad gauging station was 62,164 $m^3$ $s^{-1}$ (20.94 m) on 16 August 2019, while the peak water level was 21.16 m on 18 August with an estimated river discharge of 90,239 $m^3$ $s^{-1}$. During 2017 the measured discharge (water level) was



78,525 m³ s⁻¹ (20.79 m) on 17 August 2017, while the peak water level of 20.84 m on 16 August with estimated river discharge of 93,359 m³ s⁻¹, whereas measured and estimated maximum discharges were 102,535 m³ s⁻¹ and 103,129 m³ s⁻¹ in 1998. The peak river discharge has estimated exceedance probabilities of 1.58 %, 4.76 %, 6.35 % in 1998, 2017 and 2019 respectively (Fig. S6). The estimated annual maximum discharge in 2019 and 2017 was

lower than the one in 1998, despite higher water levels. The trend analysis of annual maximum water levels shows a positive trend at the 0.05 significance level (Fig. 10b). On the other hand, the trend in annual maximum discharge is not significant at the 0.05 significance level. Forecasting exceeding annual peak water level can provide valuable information to flood managers to take appropriate action to prevent overtopping embankments of the river.


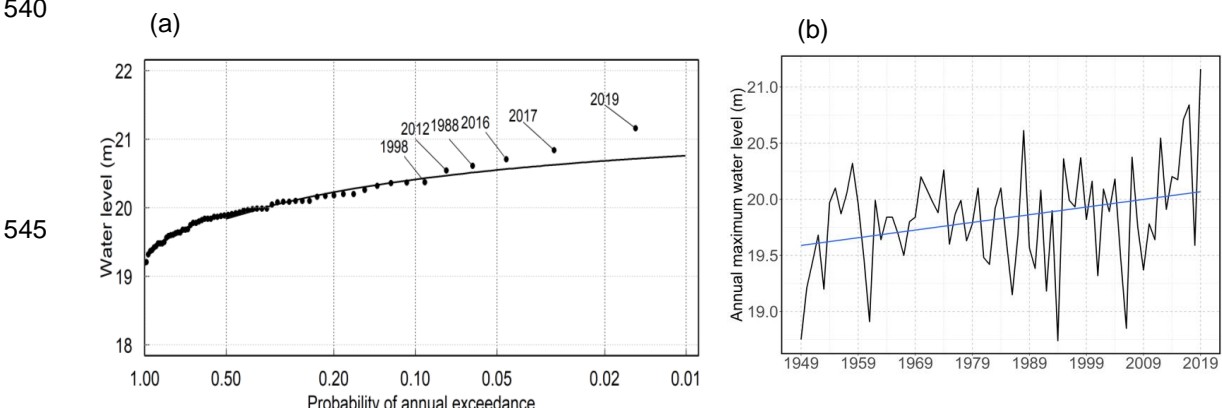

**Figure 10.** (a) Exceedance probability and (b) Trend of annual maximum water level (m) of the Brahmaputra river at Bahadurabad gauging station.

### 5.2.5 Rate of water level rise

The part of the Brahmaputra basin inside Bangladesh is a floodplain river delta and rivers usually gradually rise during floods. However, due to spatial variation of rainfall there can also be cases of a more rapid rise in water levels. The Brahmaputra river at the Bahadurabad station shows a higher rate of water level rise during the 2017 flood compared to all other years (Fig. 11a). Our investigation shows that in 2017 the river experienced a rapid rise for three consecutive days (50 cm per day) compared to two extreme years of rapid rise floods in 1988 (37 cm per

day) and 2019 (40 cm per day). The behaviour of water level rise of the tributaries was almost similar to the main course of the Brahmaputra (Fig. 11b, 11c and 11d), suggesting that the higher rate of rise in the Brahmaputra river was due to the concurrent contributions from its tributaries as result of intense rainfall on a flood-triggering hydrological sweet-spot in the lower sub-basins (Teesta, Dharla, Dudkumar). The rate of water level rise is important in order to forecast and provide timely flood warnings, as it determines how quickly the water level will

cross the flood danger level and how fast decision makers and communities need to take actions ahead of floods.



**Figure 11.** Scatter diagrams of 3-day mean water level rise (cm/day) versus water level (m) during the monsoon period in different years for the 1987–2007 period at: (a) Bahadurabad (Brahmaputra), (b) Kurigram (Dharla), (c) Pateswari (Dudkumar) and (d) Dalia (Teesta). Horizontal lines show 80th, 90th and 95th percentiles.





### 5.2.6 Flood duration

Flood duration varies annually across river networks in the basin (Fig. 12). The flood duration of the three tributaries the Dharla, the Dudkumar and the Teesta is usually shorter compared to main channel of the Brahmaputra (and most often below 16-20 days). The average flood duration in the tributaries (Teesta, Dharla, Dudkumar) is 8 days while in the main channel is 15 days. For the longest duration floods in 1998 (66 days), the Dharla (30 days) and the Dudkumar (11 days) experienced floods concurrently with the main channel while the Teesta did not flow above

danger level (Fig. 12). However, the other two long duration floods in 1988 (27 days) and 2017 (25 days) flood occurred concurrently in the main channel and all three tributaries. Flood forecast information on probable flood duration in a monsoon is essential for agricultural planning and resource management for emergency operation during floods.

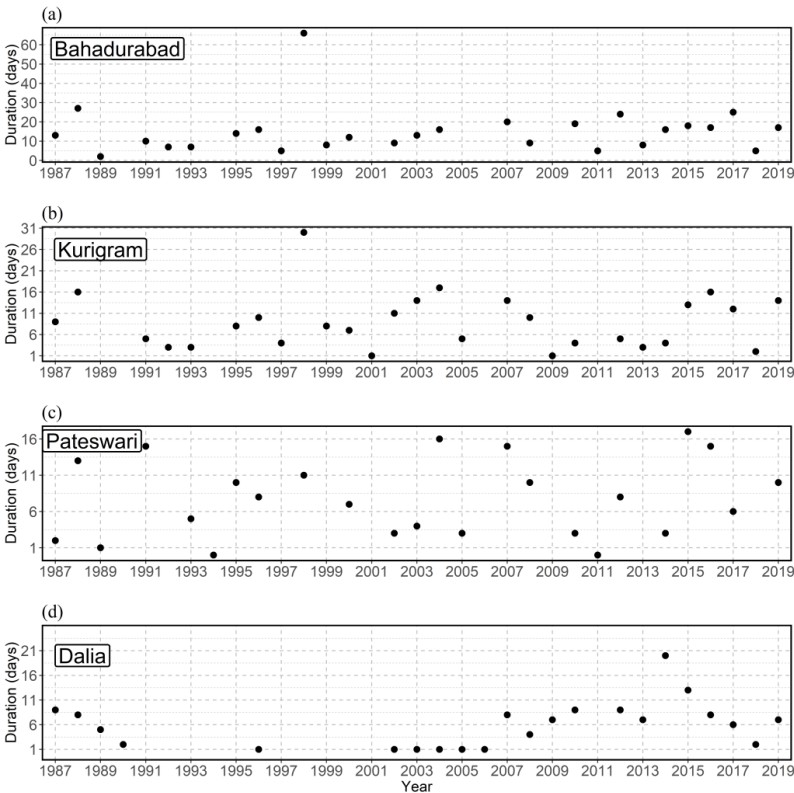

**Figure 12.** Flood duration in days above danger level from 1987 to 2019 at: (a) Bahadurabad (Brahmaputra), (b) Kurigram (Dharla), (c) Pateswari (Dudkumar) and (d) Dalia (Teesta).

615  **5.2.7 Synchronization of the Brahmaputra, Ganges and Meghna floods**

The flood characteristics of the Ganges, the Brahmaputra and the Meghna vary considerably in terms of timing, duration, and magnitude. Usually, there is a time lag between the flood peaks in the three basins, but sometimes the monsoon floods are synchronised along the three rivers. Among the three flood years analysed in detail here, only in 1998 did the streamflow of the three rivers exceeded the danger levels simultaneously from the end of





620    August to mid-September (Fig. 13a), while in 2017 the high flows were close to synchronization but there was a
time lag of 7 days between the peaks of the Brahmaputra, Ganges and Meghna (Fig. 13b). In 2019, both the
Ganges and Meghna flowed well below the danger level at the time of the Brahmaputra floods (Fig. 13c). Similarly,
there may be synchronization of the riverine floods with the high tides in the Meghna estuary: this occurred in the
1998 monsoon, when the tidal water levels were higher than average, while in 2017 and 2019 they stayed mostly

625    below average (Fig. 13d). Along with the river flow synchronization, the high spring tide at the estuary influences
the riverine flood risk creating backwater effects on the upstream river flow. Due to the long duration of the floods
in 1998, the cycle of the spring tide (usually 14 days) coincided with the flooding. Anticipation of these temporal
synchronization of floods among the three major rivers could be indicative for a potential long duration flood.

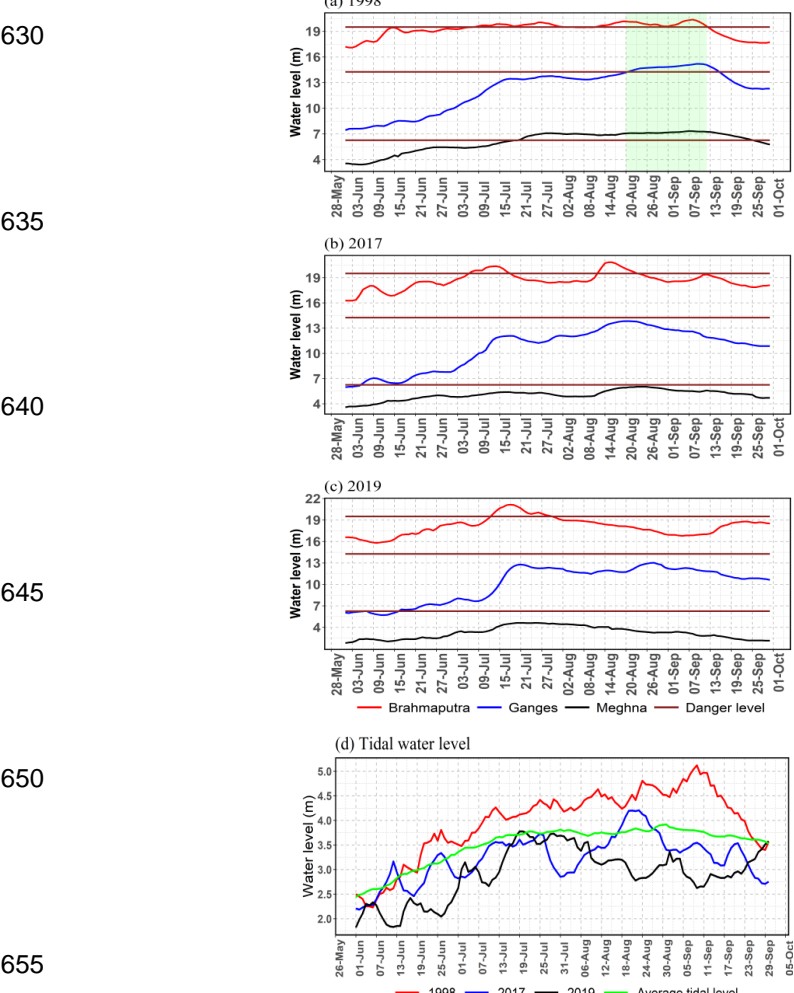

**Figure 13.** Flood hydrographs of the Brahmaputra (at Bahadurabad), Ganges (at Hardinge Bridge) and Meghna
(at Bhairab Bazar) in: (a) 1998, (b) 2017, and (c) 2019. (d) Tidal water level at Meghna estuary (Chandpur) in 1998,
2017 and 2019 along with long-term average. The shaded region in the top panel (a) shows when the three rivers

simultaneously exceeded the danger level (stations location is shown in Fig. 1b).



## 6 Discussion


### 6.1 Floods types and relevant hydrometeorological drivers

There is strong interannual variability of flood characteristics based on hydrological and meteorological drivers in the Brahmaputra basin. We structure this discussion by focussing on the top decile of events in each flood type

and considering possible caveats and counter examples; the drivers of these are summarised in Table 2. The top decile long duration floods were in 1998, 1988 and 2017; rapid rise floods were in 2017, 2019 and 1988; and high water level floods were in 2019, 2017 and 2016 (Table 2). The annual flood duration varies from a short period, i.e. two days, to more than two months. We found that the top two long duration floods (1998, 1988) were associated with basin-wide extended-range rainfall anomalies during a strong La Niña developing year. However, a strong La

Niña developing year may not always lead to the same flood characteristics; 2007 did not have exceptionally long duration floods (Table 2). While the total seasonal rainfall was higher in 2007 than 2017, the flood duration was higher in 2017 (Table 2). The rainfall varies both on seasonal as well as sub-seasonal time scales, and our results show that the basin usually receives positive seasonal rainfall anomalies during La Niña developing years. Hydrological factors such as the synchronization of floods across the Ganges-Brahmaputra-Meghna river networks,

or the influence of high spring tides at the confluence, both potentially contribute to backwater effects and slow down the receding of flood water (Mirza, 2003). There was flood synchronization among the three river basins for the long duration floods in both 1998 and 1988 (Table 2).

For the rapid rise floods in 2017, we found that localized rainfall on the sub-basins: Teesta, Dharla and Dudkumar created a concurrent rise of river flows in different tributaries leading to a rapid rise in the main channel. These

three sub-basins are located in the lower part of the Brahmaputra basin, whose confluences with the main channel are located a short distance from each other creating a "sweet spot" condition for a synchronised rapid rise. This indicates the necessity of skilfully forecasting the spatial distribution of rainfall events with sufficient lead-time to provide early warnings in Bangladesh.

Finally, while the highest water level flood in 2019 followed an extreme rainfall event with an annual exceedance

probability of 0.04%, the corresponding river discharge was not as extreme. There have been three record high water levels between 2015 and 2019, and annual maximum water levels show a significant increasing trend. However, the discharge does not follow the same increasing trend therefore other drivers such as sedimentation and other morphological changes might have played a more important role.







**Table 2.** Flood types, examples and associated key hydrometeorological conditions for the Brahmaputra river basin in Bangladesh. Data sources: [a] Hydrological database, BWDB, Bangladesh; [b] ERA5 reanalysis (C3S, 2017); [c] Bimodal tropical ISO index; IPRC/SOEST (2020); [d] Monthly SST anomalies, NOAA (2020).

| Flood Type / Key elements | Long duration | | | Rapid rise | | High water levels | |
|---|---|---|---|---|---|---|---|
| Example year | 1998 | Other extreme years 1988, 2017 | Counter example, 2007 | 2017 | Other extreme year. 2019 | 2019 | Other extreme years, 2017 2016 |
| Duration (day) | 66 days | 27 days (1988) 25 days (2017) | 20 days | 25 days | 17 days | 17 days | 25 days (2017) 14 days (2016) |
| Flood peak water level [a] (m), discharge (m³ s⁻¹) and date | 20.37 m, 1,03,129 m³ s⁻¹ (8 September 1998) | (i) 20.62 m and 98,300 m³ s⁻¹ on 31 August 1988. (ii) 20.84 m and 93,359 m³ s⁻¹ 16 August 2017 | 20.62 m, 79,779 m³ s⁻¹, 30 July 2007 | 20.84 m, 93,359 m³ s⁻¹, 16 August 2017 | 21.16 m 90,239 m³ s⁻¹, 18 July | 21.16 m 90,239 m³ s⁻¹, 18 July | (i) 84 m, 93,359 m³ s⁻¹, 16 August, 2017 (ii) 20.71 m, 28 July 2016 89,427 m³ s⁻¹ |
| Total seasonal (June to Sep.) rainfall (mm) [b] and Annual Exceedance Probability (AEP) | 1559 mm (AEP = 0.01) | 1988: 1518 mm (0.04) 2017: 1331 mm (0.55) | 1410 mm (0.29) | 1331 mm (0.55) | 1276 mm (0.71) | 1276 mm (0.71) | 2017: 1331 mm (0.55) 2016: 1249 mm (0.78) |
| Remarkable rainfall [b] events with APE and rainfall amount (mm); spatial distribution | 4 to 25 July, 1998, 0.30 (372 mm); Basin-wide  10 to 21 August 1998, 0.70 (199 mm); Localized | (i) 21 to 31 August 1988, 0.14 (239 mm); Basin-wide  (ii) 7 to 13 August 2017, 0.20 (169 mm). Localized | 13 July to 31 July 2007, 0.16, (352 mm); Basin-wide | 29 June to 12 July 2017, 0.40 (250 mm); Basin-wide  7 to 13 August 2017, 0.20 (169 mm); Localized | 04 to 16 July 2019, 0.040 (300 mm); Basin-wide | 04 to 16 July 2019, 0.040 (300 mm); Basin-wide | 29 June to 12 July 2017, 0.40 (250 mm); Basin-wide 7 to 13 August 2017, 0.20 (169 mm); Localized  13 to 25 July 2016, 0.40, (238 mm); Basin-wide |
| Departure of active days of BSISO mode [c] (phase 1 to 3) during June–Sep. from seasonal average, 25 days (+ / - indicates number of days above or below the average number of days) | -11 days | -11 days (1988) -13 days (2017) | +21 days | -13 days | -1 day | -1 day | -13 days (2017) +7 days (2016) |
| Average SST [d] anomalies (°C) Nov-Jan (NDJ) Niño3.4 region | -1.57 | 1988 (-1.85) 2017 (-0.98) | 2007 (-1.60) | -0.98 | 0.55 | 0.55 | 2017 (-0.98) 2016 (-0.56) |
| Soil moisture [b] (Seasonal, June to Sep) | Soil moisture was high (soil moisture anomaly, Fig 9) | Soil moisture in 2010 was relatively higher than 1988 and 2017, however that did not lead to long duration floods (Fig. S7) | Soil moisture was less than 1998 (Fig. S7) | Lower part of the basin is more saturated (soil moisture anomaly, Fig. 9) | Drier than 1998 and 2017 (soil moisture anomaly, Fig. 9) | Drier than 1998 and 2017 (soil moisture anomaly, Fig. 9) | Drier than 1998 and 2017 (soil moisture anomaly, Fig. 9, and Fig. S7) |
| Antecedent water level [a] (June water level) (period 1987 to 2019) | June water level was 94 cm above the historical average water level of June | June water level in all other years was lower than 1998 | June water level in all other year was lower than 1998 | June water level was 6 cm lower than historical average water level of June | June water level was 78 cm lower than historical average water level of June | June water level was 78 cm lower than historical average water level of June | June water level was 78 cm lower than historical average water level of June |
| Synchronization of floods with the Ganges and the Meghna | Floods in the Brahmaputra, the Ganges and the Meghna co-occurred between 21 August to 12 September in 1998.  The flood peak of the Brahmaputra (1,03,129 m³ s⁻¹) occurred on 8 Sep; the Ganges (74,278 m³ s⁻¹) on 09 Sep and the Meghna (10,852 m³ s⁻¹) on 7 Sep. | 1988: The common period of floods in the Brahmaputra, the Ganges and the Meghna between 24 August to 8 September in 1988. The flood peak of the Brahmaputra (98,300 m³ s⁻¹) occurred on 31 August; the Ganges (71,800 m³ s⁻¹) on 03 September and the Meghna (11,288 m³ s⁻¹) on 11 September.  2017: No flood synchronization The Ganges and the Meghna river did not flow above danger level. | The common period of foods among the Brahmaputra and the Meghna, 8 Sep. to 12 Sep. The flood peak of the Brahmaputra (79,779 m³ s⁻¹) on 30 July; the Ganges (52,013 m³ s⁻¹) on 5 August and the Meghna (10,305 m³ s⁻¹) on 5 August.  The Ganges did not flow above danger level. | No flood synchronization. The Ganges and the Meghna river did not flow above danger level. | No flood synchronization. The Ganges and the Meghna river did not flow above danger level. | No flood synchronization The Ganges and the Meghna river did not flow above danger level. | No flood synchronization The Ganges and the Meghna river did not flow above danger level. |






**6.2 Recommendations**

Our results suggest the following important scenarios for flood forecasting and disaster management in the Brahmaputra basin.

**Development of strong La Niña:** Strong La Niña development years are found to be linked with larger seasonal total rainfall; long duration floods are more likely in this scenario (50% of La Niña development years flood duration was >25 days). Therefore, a forecast of a La Niña issued at the beginning of the monsoon season (June–July) after the spring predictability barrier (when predictions of ENSO are more skilful) (Chen et al., 2020; Clarke, 2014). could provide a plausible early indication of long duration floods.

**Spatial distribution of monsoon rainfall:** The spatial distribution of the monsoon rainfall varies significantly from more localized to basin-wide, and flood responses vary accordingly in the basin inside Bangladesh. Floods can occur from two sets of rainfall distributions; basin-wide and more localized rainfall at lower sub-basins. These distributions of rainfall events give an essential scenario to forecasters about the possible rate of rise in water level. For instance, a medium-range forecast (5-10 day lead time) of a localised rainfall event over the "sweet spot" area
would indicate to forecasters that a rapid rise flood event is likely.

    **Climate change and disaster management perspectives:** Flood events with different characteristics have different challenges and impacts, and climate change is likely to influence these flood characteristics in future. Stronger interannual variability in the monsoon is expected in a warming climate (Kitoh et al., 1997; Sharmila et al., 2015), therefore for informed climate adaptation and long-term management of disaster risk, further investigation
is needed in order to understand how the two scenarios which drive interannual variability in flood characteristics might change under different climate change scenarios. It is expected that frequency of ENSO events may increase in the future under climate change conditions (Cai et al., 2014; McPhaden et al., 2020). Spatial and temporal variations of the monsoon rainfall due to climate change are important aspects that influence flood characteristics. Climate change may cause frequent extreme monsoon rainfall events by increasing the number of short-duration
rainfall events (Christensen et al., 2013; Sharmila et al., 2015; Turner and Annamalai, 2012) or changing in spatial variation of rainfall (Bhowmick et al., 2019; Christensen et al., 2013) which might lead to more frequent flood pulses with rapid rise or high water level in a monsoon.

**7. Conclusions**

This study has addressed the drivers of interannual variability of the flood characteristics that link to flood risk in the Brahmaputra basin in Bangladesh: duration, rate of rise and water level. Our results show that long duration floods are associated with seasonal and basin-wide rainfall anomalies linked to a strong La Niña development, as well the synchronization of floods with the Ganges and Meghna rivers. The rapid rate of rise in water level, which is a challenge for early warning, is driven by more localized rainfall in a hydrological "sweet spot" that causes a
concurrent contribution from the tributaries to the main channel. However, record high water levels seem to be more related to sedimentation and morphological changes than any hydrometeorological drivers.

    These drivers give a set of possible scenarios for flood forecasters to anticipate flood types during the monsoon to improve flood preparedness and risk assessment. Further study should address the potential influence of sedimentation and geomorphological changes which may have caused the recent upward trend in annual maximum
water levels in the river. In addition, given the critical link between flood duration and livelihoods in the basin, we





recommend further study of how climate change will affect the interannual variability in flood characteristics in order to support climate adaptation.

**Authors contribution**

EMS and HLC assisted with concept development. The study was led, conceived and carried out by SH who analysed the data. EMS and HLC also assisted with analysis, interpretation along with AF, AGT. All the authors contributed to the writing, reviewing & editing.

**Competing interests:** The authors declare that they have no conflict of interest.

**Acknowledgements**

This study has been carried out as part of a PhD research supported by the UK Research and Innovation (UKRI); the Natural Environment Research Council (NERC) and the Foreign, Commonwealth and Development Office (FCDO) under the Forecasts for AnTicipatory HUManitarian action (FATHUM) project (grant number NE/P000525/1) and SHEAR Studentship Cohort programme (grant number NE/R007799/1). AGT acknowledges support from the National Centre for Atmospheric Science. The authors are thankful to the Bangladesh Meteorological Department and Bangladesh Water Development Board (BWDB) for providing observed meteorological and hydrological data. We are also grateful to European Centre for Medium-Range Weather Forecasts (ECMWF) for supporting the authors as visiting scientists to carry out the research work.

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
