# Peer review of "Hydrometeorological drivers of flood characteristics in the Brahmaputra river basin in Bangladesh"

_Hydrology and Earth System Sciences, 2021_

## Author Comment (AC1)

**Responses to the comments of Reviewer 1**

We are very thankful to anonymous Reviewer 1 for his or her time to review the article and provide valuable comments. We will address these in the revised manuscript and accordingly our responses to each comment are given below. We mark our replies in blue font, while original reviewer comments are presented in black font.

**General comments**

Some parts of the manuscript require a revision to avoid repetitive ideas and improve discourse flow, as some parts of the text are rather disconnected.

Response: We will address any repetition issues in the revised version of the manuscript, as well as addressing the flow of the text.

In addition, the authors should explain better the rationale behind each of the analysis described in the manuscript.

Response: We will look to improve the rationale for each part of the analysis and make sure each part is fully justified.

Besides the structure of the manuscript, I missed a more detailed analysis of the meteorological situation leading to localised short-duration rainfall events, which are key for rapid rise floods according to the manuscript's conclusions. Have the authors considered which mesoscale or local-scale atmospheric factors contribute to these intense rainfall events? If not, are they aware of any study that analyses this aspect for the events considered here (or others of similar nature)?

Response: Localised short-duration rainfall in the region is particularly related to synoptic events such as monsoon depressions or (weaker) low-pressure systems passing from the Bay of Bengal and tending northwards across the coast. These systems are also affected by the position and strength of the monsoon trough. We have provided a short description in Section 2.2 of the movement of the monsoon trough and the role of depressions or low-pressure systems embedded within:
"Monsoon heavy rainfall in the Brahmaputra basin is associated with the movement of the eastern end of the monsoon trough to the Assam region, producing break

monsoon conditions over central India and active conditions around the Himalayan foothills (Fig. S2), often with a monsoon depression originating from the Bay of Bengal, which recurves northwards over the Brahmaputra basin i.e. Bangladesh and Assam (Dhar and Nandargi, 2003; Dhar and Nandargi, 2000; Nandargi and Dhar, 2011)."

The trough itself undergoes control by at the larger scale by the boreal summer intraseasonal oscillation (BSISO), e.g., implied by the northward movement of the trough towards the Himalayan foothills during (Indian) monsoon breaks in Krishnamurthy and Shukla (2007). We have already undertaken an analysis of the influence of BSISO events on the short-duration rainfall events which lead to flooding in 5.1.1, not finding any clear link. However, we have not undertaken any direct analysis of mesoscale features, and note from the literature that such an analysis would only likely represent such features in high resolution models e.g. Vellore et al. (2014).

In the revised version of the manuscript we will also add a short section to the discussion to consider other studies that have looked at mesoscale features, for example, noting that the interaction of orography with mesoscale features such as southward-intruding mid-latitude westerly troughs on the subtropical westerly Jetstream that can increase precipitation totals over the Brahmaputra basin (Vellore et al., 2014).

Moreover, it gives the impression that the 33-year data is not fully exploited. Some of the analyses shown in the manuscript could be extended to other years to better highlight the differences between the selected years, when exceptional floods occurred, and the rest. This is only done for some drivers, but not for all, which can be confusing for readers.

Response: While the figures in the paper largely focus on the three distinct flood types, Table 2 is used to look at additional examples and counter examples from the full 33 year record. Nevertheless, we agree that this approach could be better described, and so we will include some introductory paragraphs at the start of Section 4 to address this and justify the approach taken.

**Specific comments**

**Abstract (L26-27):** Does the last sentence refer to the drivers analysed in this study or to sedimentation and morphological changes? If it is the latter, I suggest removing this sentence, as this aspect is not investigated in this study nor anything related to climate change. In this case, I would replace this sentence by a new one that summarises the main conclusions of the manuscript.

Response: We have removed the reference to Climate Change, but don't see an issue with the comment on geomorphological changes. We show that the recent high water levels are not also observed in the discharge measurements. This means that geomorphological changes are influencing the water level, not hydrometeorological factors, and therefore addressing this is of critical importance for flood forecasting and early warning.

**Introduction:** I recommend modifying its structure. I think it would be more appropriate to start with more general aspects of flooding in the area before introducing the details regarding the most devastating cases. In addition, the whole section should be carefully revised to avoid repetitive ideas (e.g., L37-38 and L46-47) and provide a coherent text that focuses on the most relevant aspects that will be discussed in the following sections.

Response: We will revise the introduction accordingly, avoiding repetition while maintaining a logical order.

**L36:** I suggest moving Fig. S1 to the main manuscript.

Response: We will move S1 into the main manuscript.

**L59-61:** Why are these factors chosen?
"we analyse antecedent conditions (Blöschl et al., 2013; Carter and Steinschneider, 2018; Schröter et al., 2015), large-scale atmospheric and ocean anomalies (Paeth et al., 2011) and extreme statistics of precipitation and river flow (Schröter et al., 2015)."

Response: These papers that focus on different factors of flood drivers studies are relevant to the Brahmaputra flooding, because antecedent conditions, large-scale atmospheric and ocean anomalies influence on flood characteristics such as extremeness of floods. We will include this justification in our revised paper.

**L62:** Why are these cases analysed in more detail?

Response: Floods with different characteristics have different impacts, and therefore different implications for early warning and other flood management. As such it is important to understand the drivers of these different characteristic flood events to produce better forecasts of these impacts. We will add this justification to line 62.

**Section 3:** The beginning of this section is too abrupt. I suggest adding an introductory sentence indicating the value of each data source for this study.

Response: Thank you, we will add a short introductory sentence:
"This study for a transboundary river makes use of historical observed data available for inside Bangladesh, but for the wider river basin where datasets are not readily available the study is supplemented with alternative data sources such as reanalyses products. Details of each dataset are described in the following sections."

**L212:** Please, add a map illustrating the Niño 3.4 region or provide some geographical references or the area.

Response: We are reluctant to add an additional figure because of the number we have already, but we will add the following which would enable replicability for any future studies:
We use monthly SST anomaly based on the monthly Extended Reconstructed Sea Surface Temperature (Huang et al., 2017) for the Niño-3.4 region (5° N to 5° S, 170° W to 120° W) available from NOAA (2020) for the period 1987–2019.

**L236-243:** The current wording of this part of the manuscript is confusing. Please, clarify what is intended to be said.

Response: We have shortened the statement in more understandable way as given below:

"Intraseasonal variation of the monsoon wet and dry spells varies with a typical lifespan of around two weeks (Krishnamurthy and Shukla, 2007). There are two distinct modes of tropical ISOs that have been linked to this intra-seasonal variability. The first, the Madden-Julian Oscillation (MJO) is an eastward-propagating mode along the equator, whereas the second, the boreal summer intraseasonal oscillation (BSISO), also gives northward-propagating bands of convection, therefore leading to variability extending into off-equatorial monsoon regions such as South Asia (Kikuchi et al., 2012; Lee et al., 2013). While the MJO has less influence at South Asian longitudes in boreal summer, it dominates from December to April (Kikuchi et al., 2012) and hence is not considered an important driver for Brahmaputra floods, which occur between June and September."

**L250:** I propose including other years in the supplement.

Response: We will include the phase-space diagram for the BSISO during the monsoon season in other years between 1987 to 2019 in the supplementary material.

**Section 5.1:** Can mesoscale or local-scale mechanisms be analysed? (see general comments)

Response: We have responded in the general comments, above.

**Fig. 4b:** I suggest including mean JJAS rainfall for each year in this panel.
Response: We will add JJAS rainfall for each year here.

**L356:** Please, define "sweet spot".

Response: "Sweet-spot" is an informal term that has emerged from the sporting world, whereby it defines the point or area on a bat, club or racquet where it makes the most effective contact with the ball. It is now used frequently to refer to the point, location or set of conditions that would give the most effective outcome. We have checked with

colleagues that the term is translatable into Bangla, but we will adjust the sentence to be clearer:

"forming a hydrological "sweet spot" (a location where rainfall can have a more pronounced influence on river levels)"

**Section 5.1.3:** Please, add a brief introductory sentence.

Response: We will add an introductory sentence as follows:

"Next, we analysed monthly rainfall anomalies between June to September for the three flood years to understand deviations from average conditions."

**L451:** The authors should mention some of these "other drivers".

Response: Thanks, we will add further clarification:

"In addition, the recorded high-water level in 2019 is not matched by a record in the river flow, which suggests that other non-hydrometeorological drivers such as changes to the channel morphology are also important."

**L483:** The relevance of these "short-duration rainfall events" motivates an analysis of them (see general comments).

Response: We will refer back to the additional information we will add to section 5.1.2 here (see response to general comments).

**Section 5.2.3:** Please, add an introductory sentence indicating why is soil moisture relevant.

Response: We will add an introductory sentence as follows:

"Analysing soil moisture provides an indication of the importance of antecedent conditions as a driver of floods, as also addressed by Bloschl et al. (2013)."

**L488:** Can the data confirm that "basin-wide and frequent rainfall events maintained soil moisture" and thus, avoid saying "it is likely"?

Response: Our results confirm this statement, therefore we remove "likely."

**Section 5.2.4:** Again, add some introduction to this section.

Response: We will add introductory sentences as follows:
"Peak water level and discharge are two important variables needed to study annual flood characteristics. We investigated the trend and extremes of the annual peak in water level and river flow.

**L529:** I suggest including these values in a table.

Response: We will include these in Table 2.

**L534:** I recommend adding Fig. S6 to Fig. 10.

Response: We will add this in the revised version.

**L537-538:** The authors should give some recommendations to perform this task.
"lower than the one in 1998, despite higher water levels. The trend analysis of annual maximum water levels shows a positive trend at the 0.05 significance level (Fig. 10b). On the other hand, the trend in annual maximum discharge is not significant at the 0.05 significance level. Forecasting exceeding annual peak water level can provide valuable information to flood managers to take appropriate action to prevent overtopping embankments of the river."

Response: We will change the final sentence in this paragraph as follows:
"An upward trend in water levels, but not discharge, suggests that geomorphological rather than hydrometeorological changes also have a role in flood risk. To provide skilful forecasts of flooding, the Bangladesh Flood Forecasting and Warning Centre should make sure that the bathymetry information that is incorporated within the forecasting model is regularly updated."

**L563-565:** I suggest moving this sentence to the beginning of the subsection. This kind of sentences can be used as introduction for other subsections (see previous comments).

Response: Thanks, we will do this, and make similar changes to other subsections.

**L586-588:** Please, move this sentence to the beginning of the subsection.

Response: Thanks, Lines 586-588 have been moved to the beginning, with the main channel while the Teesta did not flow above danger level (Fig. 12).

**Section 6:** As it is currently written this section is more like a summary than a discussion. In addition, the two subsections include various repetitive ideas. I suggest summarising the whole section in order to succinctly provide the most important take-home messages for readers.

Response: The Discussion summarises a detailed results section into the findings that are most significant for each of the research objectives. We are happy to merge this with the conclusions into a single 'Discussion and Conclusions' section and revise the combined new section to avoid repetition.

**L671-672:** The selected cases fall in different categories. This classification should be better explained to avoid confusion.

Response: We have changed this and the previous sentence to provide a clearer explanation:
"We structure this discussion by focussing on the top decile (tenth) of events for each flood type as well as considering possible counter examples (see Table 2). The top decile flood events by duration were in 1998, 1988 and 2017; by rate of rise were in 2017, 2019 and 1988; and by water level were in 2019, 2017 and 2016."

**L674 onwards:** Strong La Niña seems to be a necessary but not sufficient condition for long duration floods. This idea should be emphasised.

Response: We will change this sentence to:

"However, a strong La Niña developing year may be a necessary but not sufficient condition for long duration floods; as an example, 2007 was not in the top decile by duration of flood."

**L686-688:** Accurate forecast of heavy rainfall is extremely challenging for small scales and long lead times. The authors should clarify the spatial and temporal scales of interest and include some references indicating the feasibility of this proposal and/or its associated challenges.

Response : We will add:

"This would require skilful forecasts of 3-day rainfall totals across areas smaller than 150 km x 150 km, but even forecasts for rainfall accumulations 1 day ahead provide poor representation of the spatial pattern of intense rainfall e.g. Hasan and Islam (2018)."

**Table 2:** I recommend moving this table to the supplement.

Response: Table 2 provides an important summary of all the analysis undertaken in the paper and is referred to several times in the text, so we believe it should be kept in the main text. We note that Reviewer 2 does not provide the same recommendation.

**Section 6.2:** Climate change and disaster management perspectives: Since climate change is not considered in this study, I suggest moving this part to the conclusions as a subject to be considered for future work.

Response: We think this recommendation will be covered by the merger of the Discussions and Conclusions and associated revisions.

**L737-738:** The simultaneity of the two factors seems crucial for the development of long duration floods. This idea should be highlighted in the discussion, rather than focusing only on strong La Niña.

Response: Lines 677-682 in the Discussion already highlight this, but we will add that La Nina also influences flows in the Ganges basin (Pervez et al., 2015).

**L746-747:** All climate change discussions should be moved here (see previous comment).

Response: This recommendation will also be covered by the merger of the Discussions and Conclusions and associated revisions.

**Technical corrections**

**L45:** Fig. 3 is cited before Fig. 2 (L115).

Response: We will adjust the ordering and make sure that figures are referenced and numbered in the order that they appear.

**L56:** I suggest "(e.g. 2017 floods). Indeed, the Bangladesh Flood Forecasting and warning Centre (FFWC) is keen to improve early warnings for these events".

Response: We will  include 2007 instead of "Indeed, the Bangladesh Flood Forecasting and warning Centre (FFWC) is keen to improve early warnings for these events "

**Caption Fig. 3:** Consider a slight modification of the caption, for instance: "Dates on which water level (WL) exceeded the danger level at the Bahadurabad station on the Brahmaputra river (see Fig. 1b for location of the gauge) indicated by a coloured dot. The colour represents WL value as indicated in the legend". Please, also indicate the threshold for severe flooding.

Response: We agree with this caption change and will use it. The severe flood threshold will also be added to the caption (20.5 m water level).

**L210:** El Ninõ (La Ninã) -> El Niño (La Niña).

Response: We will adjust the accenting as indicated.

**L215:** Please, define IPRC/SOEST.

Response: We will define IPRC/SOEST as follows:

"International Pacific Research Centre (IPRC) of School of Ocean and Earth Science and Technology (IPRC/SOEST), University of Hawaii."

References

Blöschl, G., Nester, T., Komma, J., Parajka, J., and Perdigão, R. A. P.: The June 2013 flood in the Upper Danube Basin, and comparisons with the 2002, 1954 and 1899 floods, Hydrol. Earth Syst. Sci., 17, 5197-5212, 10.5194/hess-17-5197-2013, 2013.

Carter, E., and Steinschneider, S.: Hydroclimatological Drivers of Extreme Floods on Lake Ontario, Water Resources Research, 54, 4461-4478, 10.1029/2018wr022908, 2018.

Hasan, M. A., and Islam, A. K. M. S.: Evaluation of Microphysics and Cumulus Schemes of WRF for Forecasting of Heavy Monsoon Rainfall over the Southeastern Hilly Region of Bangladesh, Pure and Applied Geophysics, 175, 4537-4566, 10.1007/s00024-018-1876-z, 2018.

Huang, B., Thorne, P. W., Banzon, V. F., Boyer, T., Chepurin, G., Lawrimore, J. H., Menne, M. J., Smith, T. M., Vose, R. S., and Zhang, H.-M.: Extended Reconstructed Sea Surface Temperature, Version 5 (ERSSTv5): Upgrades, Validations, and Intercomparisons, 30, 8179-8205, https://doi.org/10.1175/JCLI-D-16-0836.1 2017.

Kikuchi, K., Wang, B., and Kajikawa, Y.: Bimodal representation of the tropical intraseasonal oscillation, J Climate Dynamics, 38, 1989-2000, https://doi.org/10.1007/s00382-011-1159-1, 2012.

Krishnamurthy, V., and Shukla, J.: Intraseasonal and Seasonally Persisting Patterns of Indian Monsoon Rainfall, J. Clim., 20, 3-20, https://doi.org/10.1175/JCLI3981.1, 2007.

Lee, J.-Y., Wang, B., Wheeler, M. C., Fu, X., Waliser, D. E., and Kang, I.-S.: Real-time multivariate indices for the boreal summer intraseasonal oscillation over the

Asian summer monsoon region, Climate Dynamics, 40, 493-509, https://doi.org/10.1007/s00382-012-1544-4, 2013.

NOAA: El Niño Southern Oscillation (ENSO): https://origin.cpc.ncep.noaa.gov/, last access: 15/02/2019, 2020.

Paeth, H., Fink, A. H., Pohle, S., Keis, F., Mächel, H., and Samimi, C.: Meteorological characteristics and potential causes of the 2007 flood in sub-Saharan Africa, International Journal of Climatology, 31, 1908-1926, doi:10.1002/joc.2199, 2011.

Pervez, M. S., Henebry, G. M. J. N. H., and Sciences, E. S.: Spatial and seasonal responses of precipitation in the Ganges and Brahmaputra river basins to ENSO and Indian Ocean dipole modes: implications for flooding and drought, 15, 147, 2015.

Schröter, K., Kunz, M., Elmer, F., Mühr, B., and Merz, B.: What made the June 2013 flood in Germany an exceptional event? A hydro-meteorological evaluation, Hydrol. Earth Syst. Sci., 19, 309-327, 10.5194/hess-19-309-2015, 2015.

Vellore, R. K., Krishnan, R., Pendharkar, J., Choudhury, A. D., and Sabin, T. P.: On the anomalous precipitation enhancement over the Himalayan foothills during monsoon breaks, Climate Dynamics, 43, 2009-2031, 10.1007/s00382-013-2024-1, 2014.

---

## Author Comment (AC2)

**Responses to the comments of Reviewer 2**

We thank anonymous Reviewer 2 for reviewing the article and providing constructive suggestions which have improved the quality of the article. We will address these in the revised manuscript and accordingly our responses to each comment is given below. We marked our replies in blue font, while original reviewer comments are presented in black font.

Although this study would be worthy of publication in HESS, I think its current version does not meet the overall quality of HESS journal. The paper is not well written, and the structure ("key story") is not provided in a logical sequence.

Response: We will improve the structure of this detailed paper, taking on board the comments provided by both reviewers.

The idea that hydrogeological drivers (e.g., monsoon rainfall and antecedent soil moisture) determines the characteristics of floods Brahmaputra and Ganges rivers is not novel.

Response: We will endeavour to improve the clarity of our results, because this is not the idea of our paper. This study is about hydrometeorological not hydrogeological drivers, and our conclusion is not that antecedent soil moisture determines the characteristics. In addition, neither the abstract nor conclusion features any mention of antecedent soil moisture.

We believe our study is novel because it looks at the drivers of floods of different characteristics, something that has not been previously addressed. In addition, the existing literature mostly analyses the contribution of monthly or seasonal rainfall to flooding in Bangladesh, however the monsoon rainfall has strong intraseasonal variation that is reflected in the often multiple flood pulses that occur during a monsoon season.

I believe that the authors have done a lot of work analyzing the data, but authors failed to convince the readers why they are doing so.

Response: The aim is to determine the drivers of floods of different characteristics in the Brahmaputra basin, to provide information that can improve our forecasting of such floods in future. The different characteristics are important; high flood magnitude places larger numbers of the population at risk, flood duration affects livelihoods for longer, and the rate of rise increases vulnerability as people are potentially unable to evacuate in time. We will make this clearer in the revised text.

The style of this manuscript is more like "this is what we did". As a reader, I am not convinced by the authors that their findings are novel and interesting.

Response: We believe our study is novel because it looks at the drivers of floods of different characteristics, something that has not been previously addressed. In addition, the existing literature mostly analyses the contribution of monthly or seasonal rainfall to flooding in Bangladesh, however the monsoon rainfall has strong intraseasonal variation that is reflected in the often multiple flood pulses that occur during a monsoon season.
The need for this study was also driven by a positive trend in annual maximum water level in recent years which necessitated research to understand and then communicate the findings to water and disaster management authorities.

Here I provide three general (personal) suggestions for authors' review which may help improve the overall quality of this manuscript:
State the key objective or research question clearly and revise the introduction accordingly.

Response: We will revise the final paragraph of the introduction to make it clear that these are the aim and objectives. We will also provide clearer signposting at this part of the article to enable the reader to navigate their way through the analysis.

The current version of this manuscript covers, at least, two main topics: attribution of flood characteristics into hydrometeorological drivers and comprehensive analysis of three historical flood events in Brahmaputra basin. However, none of them is well defined and presented in a logical structure. If the first case is the objective, authors should increase their sample size (i.e., over 30 years flood data) to make a robust conclusion. If the second case is the goal, authors should focus contrasting three flood events (i.e., types) and highlighting the key features that cause the difference. A good example of analyzing single flood event can be found: Smith, James A., et al. "Extreme flood response: The June 2008 flooding in Iowa." Journal of Hydrometeorology 14.6 (2013): 1810-1825.

Response: Thank you for your comments, we agree that the aims and objectives should be more clearly defined, and this should help address the lack of clarity here. We don't necessarily agree that the two topics are different: we look at the most extreme of three different flood types and draw a hypothesis as to the drivers of each type of flood. We then use the wider record to determine whether other flood events support these hypotheses. There are obviously limitations when only using a 33 year record, but we are limited by the length of the available hydrological record in Bangladesh.
Thank you for suggesting that reference, this should have been included in the introduction but was overlooked.

Based on the determined research objective, authors should consider remove some unnecessary analyses which fail to directly support the main conclusion. The current study used GEV distribution, trend analysis, correlation between climate indices with floods, wavelet transform etc. However, some of the analysis does not directly support the conclusion. For example, authors show there is a trend in water level but failed to attribute this trend to any of the hydrometeorological drivers and to explain how this trend affects the flood characteristics in general. If authors want to include a conclusion or result, defend it in detail. Otherwise, drop it.

Response: We include analyses when they directly respond to the aims and objectives, we wouldn't only include results which supported the main conclusion because it is important to understand and communicate the limitations of our findings.

Failing to find any hydrometeorological driver for the trend in water level is a key finding, not one that should be 'dropped', because an upward trend in water level affects millions of lives. Highlighting that further scientific research (and data collection) is needed to understand the cause of this upward trend is an important message for the scientific community.

Go through the paper and make every sentence convincing and logical. Also, delete the sentences conveying the same idea. As a reader, some sentence sound vague and does not provide the information I am expected to understand. Here are two examples:

Line 556: "*However, due to spatial variation of rainfall there can also be cases of a more rapid rise in water levels.*" I am expected to understand the reason why rainfall heterogeneity causes the rapid rise in water levels at the gage. Is it because rainfall hit the region where the watershed slope is high? Are these rainfall have similar temporal distribution?

Response: We agree that this paragraph could be better written. We will change it to: "The part of the Brahmaputra basin inside Bangladesh is a floodplain river delta and rivers usually gradually rise during floods. The Brahmaputra river at the Bahadurabad station shows a higher rate of water level rise during the 2017 flood compared to all other years (Fig. 11a), something which caught FFWC by surprise (FFWC, personal communication). In 2017, the river experienced a rapid rise for three consecutive days (50 cm per day) compared to two extreme years of rapid rise floods in 1988 (37 cm per day) and 2019 (40 cm per day). The timing of the water level rise in the lower sub-basins of Dharla, Dudkumar and Teesta tributaries was almost similar to the main course of the Brahmaputra, suggesting that the high rate of rise in the Brahmaputra river was due to the spatial pattern of rainfall over these three tributaries (Figure 7b) on a flood-triggering hydrological sweet spot. The rate of water level rise is important in order to forecast and provide timely flood warnings, as it determines how quickly the water level will cross the flood danger level and how fast decision makers and communities need to take actions ahead of floods. However, due to spatial variation of rainfall there can also be cases of a more rapid rise in water levels."

Line 534: *"The estimated annual maximum discharge in 2019 and 2017 was lower than the one in 1998, despite higher water levels"* In most gages, the discharge is estimated using rating curve and water level. So, I am expected to understand why high water level is linked to a low discharge, which authors failed to provide.

Response: We don't believe we have failed to provide this. Section 3.1 provides a description of the hydrological observed data used in this study, and Lines 191 to 192 in particular state that river flow is measured using 'a current meter (or Acoustic Doppler Current profiler)', in addition to estimates of river flow from a rating curve. In the Results section 5.2.4 details where the flows used are the measured flows or the estimated (rating curve) flows.

We will include the following text to explain why there can be higher water level / lower discharge:

The Brahmaputra is a braided river which undergoes changes each year due to erosion and sedimentation. The Bangladesh Water Development Board (BWDB) has recalibrated the compound rating curve from three segments to two segments since the 2007 monsoon season due to morphological changes such as decreases in river width and depth at the gauging station (personal communication with the hydrologist, BWDB). Therefore, as the channel capacity has got smaller, the same discharge would lead to higher water level.